



# Performance of GPM-IMERG precipitation products under diverse topographical features and multiple-intensity rainfall in an arid region

Safa A. Mohammed[1], Mohamed A. Hamouda[1,2], Mohammed T. Mahmoud [1,3], and Mohamed M. Mohamed [1,2]

[1]Department of Civil and Environmental Engineering, United Arab Emirates University, Al Ain, Abu Dhabi, United Arab Emirates
[2]National Water Centre, United Arab Emirates University, Al Ain, Abu Dhabi, United Arab Emirates
[3] Civil Engineering Department, University of Khartoum, Khartoum, Sudan.

*Correspondence to*: Mohamed A. Hamouda (m.hamouda@uaeu.ac.ae)

**Abstract.** The influence of topographical features and rainfall intensity on the accuracy of precipitation values estimated by earth observing satellites has attracted attention in the past decade. Assessment of rainfall products delivered by the Integrated Multi-satellitE Retrievals of Global precipitation measurement (IMERG) against ground observations has risen as an important endeavour since the accuracy of these products remain unreliable. This study comprehensively evaluated the three GPM IMERG products (near and post-real-time), over the period March 2014 to June 2018. The evaluation approaches were carried out for different seasons, rainfall intensities, topographical features, and hydrological regions over an extremely arid and semiarid country of Saudi Arabia. In general, the results confirmed that the performance of the final-run product surpassed the near-real-time products in terms of consistency and estimated errors. The evaluation results showed that for seasonal-based evaluation, the precipitation products exhibited better performance in spring and summer, while having relatively lower accuracy and higher biases in fall and winter. In addition, the results showed that the IMERG products had high performance in capturing the various rainfall intensities, with light rain having the highest accuracy. This is particularly important for arid regions as most of the rainfall is of the low-intensity class. Overall, the higher the rainfall intensity, the higher the detection errors in the IMERG products. Moreover, the hydrological evaluation results showed that the hydrological regions with low density of rain gauge stations hinders the proper evaluation of satellite products and tends to underestimate the performance of the products. Furthermore, the accuracy of the precipitation products was affected by topography to different extents. IMERG precipitation products exhibited high detection accuracy over moderate elevation areas (inland regions); whereas it had poor performance over flat plains (coastal regions) and high altitudes (foothills and mountainous regions). The outcomes of this evaluation could help developers in improving the GPM IMERG calibration to achieve better detection accuracy over arid and semiarid regions. More importantly, these results are of interest for local authorities to help manage development activities and to plan precautionary measures for extreme rainfall events.



# 1 Introduction

Precipitation is one of the most complex natural processes in the hydrological cycle that undergoes momentous variability at spatial and temporal scales. The acquisition of accurate precipitation measurements is crucial since it is the main input in a

wide range of applications such as climate change prediction, environmental studies, hydrological modelling, flood forecasting, drought monitoring, and water resources assessment. In addition, precipitation measurements at a high spatial and temporal resolution are crucial to properly simulate the hydrological states of natural systems. Precipitation characteristics, such as the rainfall pattern, intensity, probability distribution of rainfalls and return periods, are considered the basis for studying the hydrological behavior of any catchment (Derin and Yilmaz, 2014; Li et al., 2013; Mahmoud et al., 2018; Wu et al., 2012).

The main sources of rainfall data are measurements of rain gauge stations, observations of weather radar, and satellite-based rainfall measurements (Zhao et al., 2017). In most of the cases, rain gauge networks are poorly distributed, particularly in mountainous topographies. Thus, records of rain gauge stations may not reflect the hydro-metrological characteristics of such a region accurately (Eris and Agiralioglu, 2012). Many researchers tried to apply interpolation techniques to estimate rainfall for poorly gauged and ungauged regions. However, several drawbacks from using these techniques were reported due to

technical limitations such as insufficient rain gauge stations and poor quality of data (Eris and Agiralioglu, 2012). In addition, most rain gauge stations have numerous measurement problems, for example, observer errors, evaporation of precipitation, and errors of heavy rain measurements due to splashing. Weather radars provide measurements that are with high spatial resolution over a certain region and can be easily achievable compared to developing networks of rain gauge stations. Radars, however, have deficiencies due to poor coverage and problems such as beam blockage in complex topographic features (Furl

et al., 2015). On the other hand, satellite-based precipitation estimates offer timely, quasi-global coverage, and are not inhibited by local topography. Furthermore, a lot of effort has been put into the validation and verification of satellite-based perception estimates (Mantas et al., 2015; Nicholson et al., 2003). Nevertheless, sub-par performances of satellite estimates were commonly reported globally, particularly with their ability to precisely capture high rainfall magnitudes. The main superiority of satellite data over rain gauge data is that it provides uniform spatial coverage at high temporal resolution. Moreover, Satellite

products are valuable in mountainous regions as it provides well coverage in comparison to both weather radar observations and gauges data (Boushaki et al., 2009). However, it is sometimes difficult to assess their accuracy due to the lack of reliable ground-based observations (Hirpa et al., 2010).

In the last three decades, meteorologists and hydrologists were attracted by the advancement in the satellite information technology resulting in developing algorithms to retrieve precipitation data from cloud information. These algorithms estimate

precipitation amounts from the characteristics of clouds as interpreted from infrared (IR), visible (VI), and microwave (MW) satellite images (Boushaki et al., 2009). Generally, passive microwave (PMW) measurements have demonstrated high performance on the global scale compared to the algorithms based on IR and VI while precipitation estimations that are based on IR have a higher temporal resolution than others (Ebert et al., 2007; Hobouchian et al., 2017). Currently, most of the recognized satellite-based precipitation products are a different combination of MW and IR to benefit from their



complementary strengths. For instance, the (PERSIANN) method which refers to Precipitation Estimation from Remotely
Sensed Information using Artificial Neural Networks generates rainfall estimates by applying driving relations between IR
and MW data to IR data (Sorooshian et al., 2000). Furthermore, other methods such as Climate Prediction Center Morphing
Method CMORPH  generates a rainfall estimates that are derived from MW data while IR data, that used to propagate the rain
pixels by a tracking approach to derive a cloud's motion field (Joyce et al., 2004).

The first devoted meteorological precipitation satellite is the Multi-satellite Precipitation Analysis (TMPA) 3B42R that was
produced by the Tropical Rainfall Measuring Mission (TRMM). The National Aeronautics and Space Administration (NASA)
launched the successful satellite in late 1997. The TRMM products were used widely, and the recent update was released in
2012 including two products: the post-processed product (3B42-V7) and the near real-time product (3B42RT). The TMPA
applies an estimation method that relies on using IR calibrated estimates with MW estimates. The TMPA product provides

estimations of tropical precipitation with good accuracy (Hirpa et al., 2010; Huffman and Bolvin, 2007; Huffman et al., 2007;
Prakash et al., 2018). Recently, in 2014, NASA and Japan Aerospace Exploration Agency (JAXA) cooperated to launch the
Global Precipitation Measurement (GPM) satellite, after the impressive success of TRMM. It consists of one main observatory
satellite and ten other partner satellites, carrying a up-to-date Dual-frequency Precipitation Radar (DPR), GPM Microwave
Imager (GMI), and other innovative instruments (Huffman and Bolvin, 2007; Kim et al., 2017). The satellite is anticipated to

perform efficiently in the prediction of flood hazards and reduce uncertainties in estimating short-term precipitation as it has
a high spatiotemporal resolution (De Coning and Poolman, 2011; Sharifi et al., 2016).  According to NASA, the GPM provides
four levels of data; which are Level-0, Level-1, Level-2 and Level-3. The concerned product that is representing Level-3, is
the Integrated Multi-satellitE Retrievals for GPM (IMERG), released in the early 2015, and has since gained more attention
and recommendations from researchers and practitioners. IMERG products have a high resolution (spatially 0.1° latitude ×

0.1° longitude) and multiple temporal resolutions (ranging from half-hourly, up to monthly basis). It includes three modes of
output namely, early, late, and final (the post-real-time) runs based on latency and accuracy. Its data incorporate, merge and
calibrate many features from TMPA, CMORPH, and PERSIANN precipitation products (Wang et al., 2017).

While the recent advancement in satellite precipitation products promises a parallel advancements in related applications in
the fields of meteorology and hydrology, it is important to note that there is a continuous  need for the investigation and critical

evaluation of their products' accuracy and performance in estimating precipitation events under varied conditions (Furl et al.,
2018).  Generally, satellite-based precipitation products provide more accurate estimates in tropical wet and dry zones than in
mountainous and semi-arid regions (Thiemig et al., 2012). Several studies indicated that the Tropical Rainfall Measuring
Mission (TRMM) and its continuation mission GPM provide more accurate and relatively consistent estimates. However, it
could underestimate heavy rainfall events and overestimate average rainfall events (Omranian and Sharif, 2018; Vernimmen

et al., 2012).

One of the often-discussed challenges for satellite precipitation retrievals is areas with complex topography where the
precipitation has high spatiotemporal variation (Hobouchian et al., 2017). It is not common to find rain gauges in mountainous
regions due to accessibility issues. In addition, since most developments exist in lowlands, it follows that most of the rain





gauges are concentrated in lowlands while the highlands are left under-represented. Since this under-representation can be
augmented with satellite products, researchers have started focusing on evaluating satellite precipitation products over complex
topography (Blacutt et al., 2015; Dinku et al., 2010; Gao and Liu, 2013; Habib et al., 2014; Hirpa et al., 2010; Salio et al.,
2015). These efforts represent a good start, but more research is needed to cover different topographic and climatic regions
around the globe. Results show that the main sources of error in the satellite precipitation measurement are from IR and PMW
retrievals. The thermal thresholds of IR retrievals, which are used to differentiate between the rainy and unrainy clouds, will
underestimate the heavy rainfall events (Dinku et al., 2008, 2010) and miss the light rain (Hong et al., 2007) resulting from
warm orographic clouds over mountainous regions. Whereas, PMW retrievals depend on the ice loft to detect the heavy rainfall,
where this is not the situation in the mountainous regions as heavy rain events could happen without ice particles due to the
warm orographic clouds (Dinku et al., 2010). In some cases, PMW retrievals overestimate precipitation as they are misled by
the ice covers and cold atmosphere on the top of the mountains and consider them as rainy clouds (Derin and Yilmaz, 2014;
Hirpa et al., 2010; Hobouchian et al., 2017)

In addition to complex topography, low rainfall intensity events represent another challenge for satellite precipitation products.
Although light rain covers vast areas of the globe, particularly the subtropics, yet to-date there are limited studies attempting
to evaluate the accuracy of satellite products in light rain detection. It has been observed that the trends of light rain decreased
along the period of 1973 to 2009 in Asia, North America, and Europe (Qian et al., 2010; Song et al., 2017). Light rain can be
classified into two types, either intense showers falling over small areas or light rain falling over large areas (e.g. drizzles).
PMW sensors fail to detect these intense showers or light precipitation that happen over only a few kilometers since its footprint
resolution ranges between 10 to 50 km (Song et al., 2017). TRMM Precipitation Radar TRMM PR is only sensitive to the
precipitation over 0.5 mm/h. One of the missions of the GPM is to improve the monitoring of snowfall and light rain by using
highly sensitive PMW sensors and taking higher samples (Hou et al., 2014). GPM has a passive microwave radiometer that
includes 13 channels called GPM Microwave Imager (GMI) coupled with Ka/Ku-band dual-frequency precipitation radar
(DPR) (Draper et al., 2015). The sensitivity of the latter is 0.2 mm/h which is higher than that of the TRMM precipitation radar
promising to detect light rain and snowfall at high latitudes (Prakash et al., 2018).

The Kingdom of Saudi Arabia (KSA), is a relatively large (2.15 million km2) country that mostly falls within the same climatic
zone, attracted several studies for evaluating satellite-based precipitation products. Studies that evaluated TRMM products
over Saudi Arabia discovered that there was considerable variation in the accuracy of the products for different events and
sub-regions; the conclusion was that TRMM cannot be the only source of data for hydrological as it provides limited input
information (Almazroui, 2011; Kim et al., 2017). Another study validated the IMERG products throughout Saudi Arabia, the
findings reflected that the final-run product had a greater performance in detecting and estimating the precipitation over large
portion of the focused area (Mahmoud et al., 2018). This study was, however, limited to only two seasons (winter and fall
2015-2016), which calls for a more comprehensive study over a longer period.

This study performed to assess the accuracy of GPM IMERG three products namely; early, late and final run products, over
Saudi Arabia. The evaluation was conducted by using ground observation data acquired from 275 rain gauge stations at daily





scale over the period March 2014 to June 2018. The evaluation of IMERG satellite products in this study will be threefold: (1) evaluate the impact of spatial characteristics (topographical and hydrological zones) on IMERG products; (2) evaluate the
variation in IMERG performance in different seasons; and (3) evaluate the performance of IMERG in detecting rainfall at different intensities. Overall, this study presents a comprehensive evaluation that considers a wide range of evaluation approaches on both temporal and spatial scales. More importantly, for the first time in the study area, this study evaluates the IMERG precipitation products performance under the five rainfall intensity categories (light, moderate, heavy, storm and strong storm) with a focus on light rain detection.

**2 Study Area**

The study is focused on The Kingdom of Saudi Arabia (KSA) that occupies an area of about 2,250,000 km$^2$, which is just under eighty percent of the Arabian Peninsula. The country covers a complex topographical surface, which falls between 43°-55°E and 32°-16°N, as represented in Fig. 1. Its vast area (with a wide latitude expanse) combined with its topographical variation resulted in diverse precipitation over the area (Hasanean and Almazroui, 2015; Mahmoud et al., 2018). The country
has thirteen administrative regions and about 400 rainfall stations (Al-Zahrani and Husain, 1998).

Although the country has a wide climatic range due to spatial and temporal temperature variability, it is considered one of the driest countries in the world. In addition, the eastern and southeastern regions of KSA include the largest sand desert in the world, called The Rub Al-Khali (Empty Quarter) (Hasanean and Almazroui, 2015). Based on the aridity index, as defined by (Topographic, seasonal and aridity influences on rainfall variability in western Saudi Arabia), the majority of the country's
area is classified as a desert climate, where precipitation is infrequent, and temperatures are high, with an exception of the mountainous region as it is considered a semiarid region. The main factors behind the peculiar climate of KSA are its sub-tropical latitude and location. It is sandwiched between the massive continental land of Africa and Asia, and at the same time close to the circum-global latitudinal belt, that has high atmospheric pressure. These factors make KSA one of the hottest and lowest humidity countries in the world, except for its coastal lands (Hasanean and Almazroui, 2015; Horton et al., 2010).

In general, precipitation over KSA is variable with around 100 mm total annual rainfall. In the northern half of the country, the rainy season starts in October and ends by April while there is almost no precipitation for the remainder of the year. Rain in this area results from the feeble weather originating from the Mediterranean or North Africa. The southwestern region, on the other hand, experiences a different precipitation pattern resulting from a mountain range that extends to western Yemen in a north-south orientation along the Red Sea, with heights more than 1500 m. These mountains cause an uplifting of the Indian
monsoon and the occurrence of heavy rainfall in the region. Overall, this part of KSA is characterized by rainfall during the whole year due to convective rain driven by topography (Hasanean and Almazroui, 2015; Horton et al., 2010).

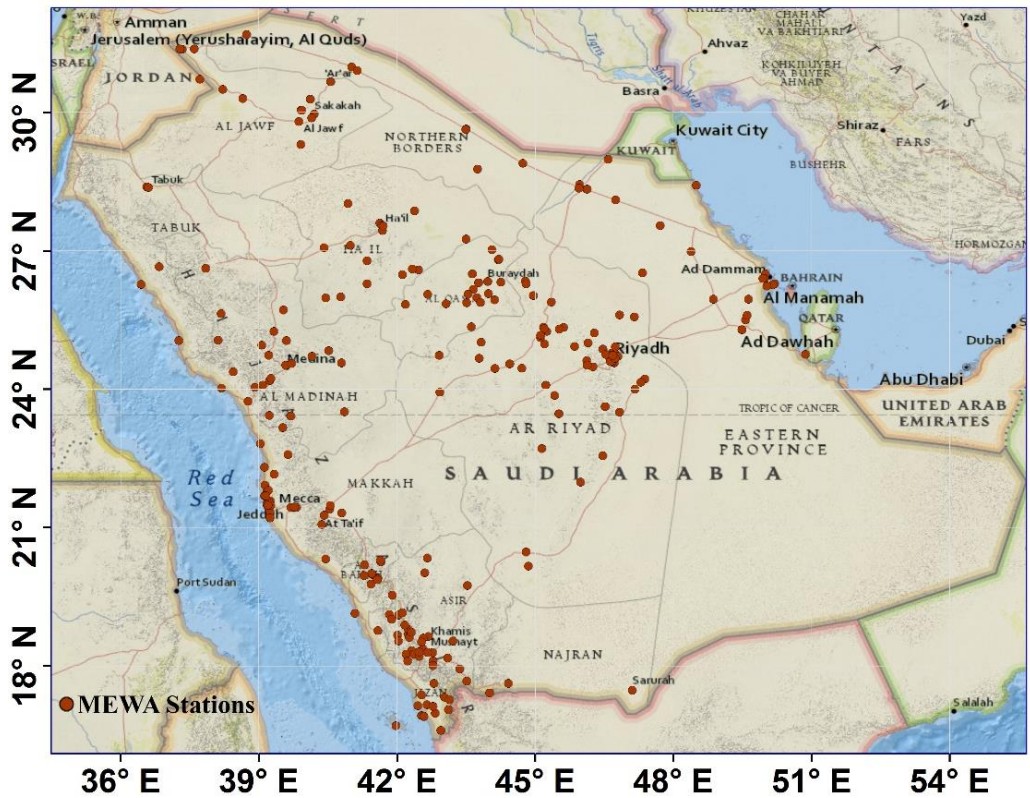

**Figure 1.** Distribution of rain gauge network operated by Ministry of Environment, Water, and Agriculture (MEWA), KSA.

## 3 Rainfall Datasets

### 3.1 Rain Gauge Dataset

This study intends to validate the accuracy of the satellite precipitation estimates over the whole KSA, which has different topographical and hydrological zones. Ideally, well-distributed and sufficiently dense rain gauge stations should be used for such a study. Many researchers used the GPCC (http://www.dwd.de/) gauge data for evaluation purposes; this data is not of sufficient density nor distribution for obtaining ground observation data at fine spatial resolution (Wang et al., 2017). In this study, we have used daily rainfall data obtained from the Ministry of Environment, Water, and Agriculture (MEWA) of Saudi Arabia. The data includes more than 270 gauges distributed over the country (see Fig. 1), and extending for the period from March 2014 to June 2018 to match the GPM Satellite data availability. These MEWA rain gauge stations are not included in the GPCC data; therefore, the evaluation carried out in this study is independent of the calibrated satellite data especially since the IMERG final product is only calibrated using GPCC data. Overall, the data covers most of the study area with a slight variation in distribution. It can be observed that the rain gauge stations are concentrated in the western and southwestern parts of the country as well as the middle part, which has the country's capital city Riyadh. The eastern part also has a good



distribution of the gauges. On the other hand, the northern part and the southeastern parts of the country have a sparse distribution of rain gauge stations.

## 3.2 IMERG Dataset

The IMERG data is provided at a spatial scale of approximately 11 × 11 km and between 60°S - 60°N with different temporal scales. In this study, we have used the finest temporal resolution (half-hourly) data. It includes three modes of runs, near real-time: early-run (IMERG-E) and late-run (IMERG-L), and post-real-time: final-run (IMERG-F). The differences between the three products are the time of release and the calibration process. The near-real-time products are pure satellite products, which are released 4 hours and 12 hours after a real-time, respectively; while the post-real-time IMERG-F is calibrated with the

GPCC data and released after about 2 months. The IMERG products were requested and collected from NASA's website through the link (https://pmm.nasa.gov/data-access/downloads/gpm).

## 4 Methodology

The primary objective of this study is to assess the capability of the IMERG products for detecting the rainfall under low intensity, and over various topography over the KSA. Rainfall ground observations during the period from March 2014 to June

2018 were used as a reference in this study. The evaluation was carried out in the following main steps: preparation of rain gauge data, processing of the satellite data, and performing the spatiotemporal evaluation of the GPM satellite data versus the gauged-based data using a set of statistical indices.

### 4.1 Data Preparation and Processing

Precipitation data was downloaded from MEWA website for all KSA rain gauges that observed a rainfall during the study

period. The data was stored in a database containing the longitude, latitude, altitude, and the UTM zone of each station to use in subsequent steps. The data was provided with Hijri dates and was converted to Gregorian date to be able to match it with the IMERG data. Regarding the GPM IMERG data, a script written in R was used to extract data for the study area, as the GPM data covers almost the entire globe. The raw GPM IMERG data provides each half-hourly data in one file. Therefore, the number of processed files to cover the study period was 75,480 for the period from March 2014 to June 2018. In addition,

the data was adjusted to match KSA day, as the day begins 3 hours ahead of GMT. Since the reference rainfall (gauge-based) data was available at the daily temporal resolution, the IMERG data was aggregated from half-hourly to daily to sustain the homogeneity in the comparison.

The next step of the analysis was to determine and compile the rainfall events happening during the study period based on ground observations. In this study, daily precipitation values that amounted to zero was discarded and values greater than zero

were considered as 'rainfall observation'. The next step in the analysis was to determine the GPM grids points representing rain gauge stations (point to point analysis). An algorithm (a script) was used to select the accurate grid point in the IMERG


file that matches the coordinates of the ground station. Based on the event date and UTM zone, the developed code would select and read the corresponding IMERG processed file and take the nearest intersection of grids to the ground station coordinates. This extracted value was then compared to the value obtained by the rain gauge station. Statistical performance

measures were calculated to evaluate the accuracy of the IMERG products. The aforementioned steps were coded to ease and automate the analysis process. Fig. 2 demonstrates the structure of each module and how each part of the analysis was implemented.

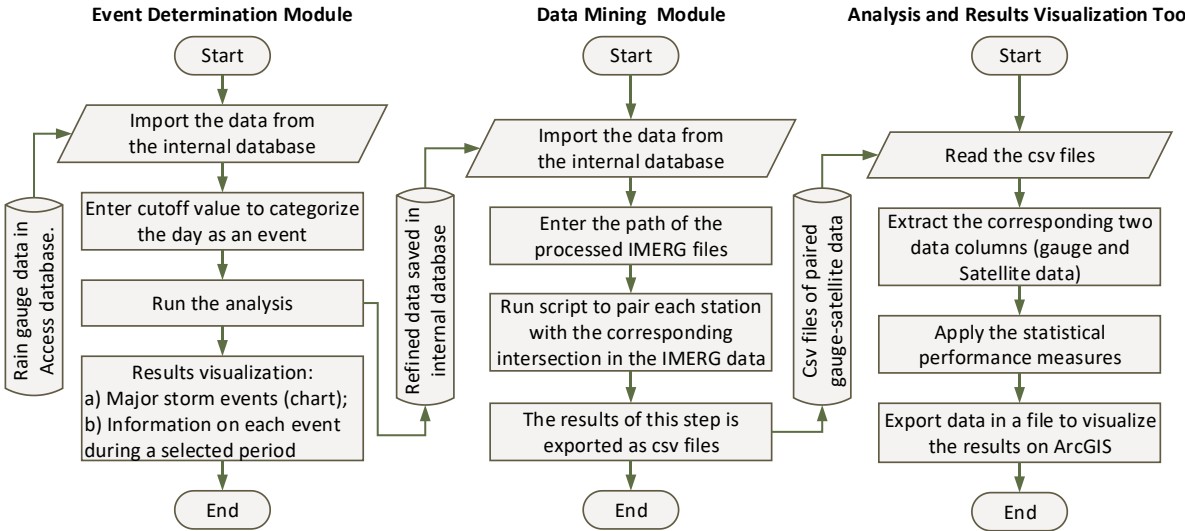

**Figure 2.** Schematic diagram of the event determination, matching coordinates, and data analysis modules.

## 4.2 Statistical Evaluation Indices

Quantitative statistical indices were used to evaluate the accuracy of the GPM IMERG products against the ground station observations. In this study, statistical measures were divided into three main groups: the categorical statistical indices, the classical statistical indices, and the correlation coefficient index. The first group of indices describes the detection accuracy of the IMERG measurements. It includes the probability of detection (POD), which measures the ratio of ground observations

that were correctly detected by the IMERG estimates, with an optimum value equal to 1; and the critical success index (CSI), which also describes the ability of IMERG products to detect the precipitation over an area. In this study, since the data analysis and the evaluation process are based on the events and not on the whole time series, both POD and CSI will give same values as there will be no precipitation events detected only by satellite. The second group measures the level of error and bias in IMERG products; it includes mean absolute error (MAE), root mean square error (RMSE) and relative bias (RBIAS). MAE

provides a general assessment of the errors of IMERG precipitation data versus rain gauge observations while RMSE quantifies the magnitude of these errors. On the other hand, RBIAS shows the systematic errors between the two precipitation data sets. The third group indicates the consistency of the IMERG products with ground observations which is represented by the





correlation coefficient (CC) that provides a measure of agreement between IMERG products and ground observations. The following formulas used to calculate the statistical indices.

**Table 1.** List of statistical evaluation indices utilized to assess the IMERG-E, IMERG-L, and IMERG-F

| Statistical Indices | Formulae | Optimum Value |
|---|---|---|
| Probability of Detection (POD) | $POD = \dfrac{N_{OD}}{N_{OD} + N_O}$ | 1 |
| Critical Success Index (CSI) | $CSI = \dfrac{N_{OD}}{N_{OD} + N_D + N_O}$ | 1 |
| Mean Absolute Error (MAE) | $MAE = \dfrac{1}{n} \sum_{i=1}^{n} \lvert D_i - O_i \rvert$ | 0 |
| Root Mean Square Error (RMSE) | $RMSE = \sqrt{\dfrac{1}{n} \sum_{i=1}^{n} (D_i - O_i)^2}$ | 0 |
| Relative Bias (RBIAS) | $RBIAS = \dfrac{\frac{1}{n}\sum_{i=1}^{n}(D_i - O_i)}{\sum_{i=1}^{n} O_i} \times 100\%$ | 0% |
| Correlation Coefficient (CC) | $CC = \dfrac{\sum_{i=1}^{n}(D_i - \bar{D})(O_i - \bar{O})}{\sqrt{\sum_{i=1}^{n}(D_i - \bar{D}) \sum_{i=1}^{n}(O_i - \bar{O})}}$ | 1 |

Where $n$ is number of records, $D_i$ is detected precipitation value by the satellite, $O_i$ is observed rainfall value by ground stations, $\bar{D}$, $\bar{O}$ are mean values of $D_i$ and $O_i$, $N_{OD}$ is the number of observed and detected events by both satellite and rain gauge , $N_O$ is the number of events that are observed by rain gauge but not detected by the satellite, $N_D$ is the number of events that are not observed by the rain gauge but detected by the satellite.

**4.3 Evaluation Techniques**

In this study, the evaluation of the accuracy of GPM IMERG precipitation products was divided into three main areas, namely: temporal evaluation, spatial evaluation (topographical and hydrologic), and rainfall intensity-based evaluation.

**4.3.1 Seasonal Evaluation**

The seasonal evaluation was conducted by analyzing rainfall on a seasonal basis throughout the study period over the entire
study area. Ground observations and IMERG datasets were prepared and grouped to represent the four seasons of KSA which are: December to February (winter), March to May (spring), June to August (summer), and September to November (fall) as stated by (Hasanean and Almazroui, 2015). The evaluation focused on investigating the capability of IMERG products to accurately detect the rainfall in the season and determine the rainfall distribution variability within the season. The objective of this analysis is to verify the utility of IMERG products as a source of data that could help bring a better understanding of
the climate of Saudi Arabia. This analysis could also highlight the use of GPM data in monitoring climate change in the region,




particularly with respect to forecasting possible future changes of both potentially hazardous events and recurring seasonal events. This information is important for development sectors, such as agriculture and tourism. In order to contextualize these potential future changes, this evaluation of current and recent time periods needs to be well understood both in terms of seasonal phenomena and extremes (Horton et al., 2010).

### 250 4.3.2 Rainfall Intensity-Based Evaluation

The objective of this evaluation was to explore the ability of IMERG products in detecting the different precipitation intensities ranging from light rain to large storms events. The focus was on the detection of light rain since it is the most frequent type of precipitation in arid regions such as Saudi Arabia. The evaluation was neither strictly temporal nor spatial, but rather a combination of both. It evaluates the detectability of events that had fallen over the entire study area, and throughout the entire 255 study period categorized only by precipitation intensity. The approach required imposing various thresholds to classify precipitation intensities thresholds were adopted from Chinese Meteorological department (Zheng et al., 2014) (Table 2).

**Table 2.** Rainfall classification according to the Chinese meteorological department (Zheng et al., 2014)

| Rainfall Class | 24h Rainfall Amount (mm) |
|---|---|
| Light rainfall | < 10 |
| Moderate Rainfall | 10-25 |
| Heavy Rainfall | 25-50 |
| Storm | 50-100 |
| Large Storm | 100-250 |
| Extreme Large Storm | ≥ 250 |

### 4.3.3 Spatial Evaluation

The spatial evaluation was conducted based on two major approaches: topographical and hydrological zoning of the study 260 area. The first approach was a topographical-based evaluation, carried out in order to assess the influence of topography on the performance of the IMERG precipitation measurements. This is particularly important since Saudi Arabia has a complex topography. The study area was divided into five topographical regions according to (Elnesr et al., 2010) as shown in Fig. 3. The topography in KSA varies from low altitudes in the coastal areas (0 up to 100m) to high altitudes in the mountainous areas (more than 2000m). The second approach was a hydrological-based evaluation. The KSA is divided into ten hydrological 265 regions with unique characteristics, as discussed in previous studies (Al-Zahrani and Husain, 1998). The study area was divided into the 10 hydrological regions, each of which including several ground stations. The objective of this approach was to test whether or not the IMERG products can perform accurately in the different hydrological regions. The results can be of interest to government authorities conducting development plans in the different hydrological regions. Fig. 4 shows the hydrological regions of Saudi Arabia based on the Ministry of Agriculture and Water map (1984) (Elnesr et al., 2010).

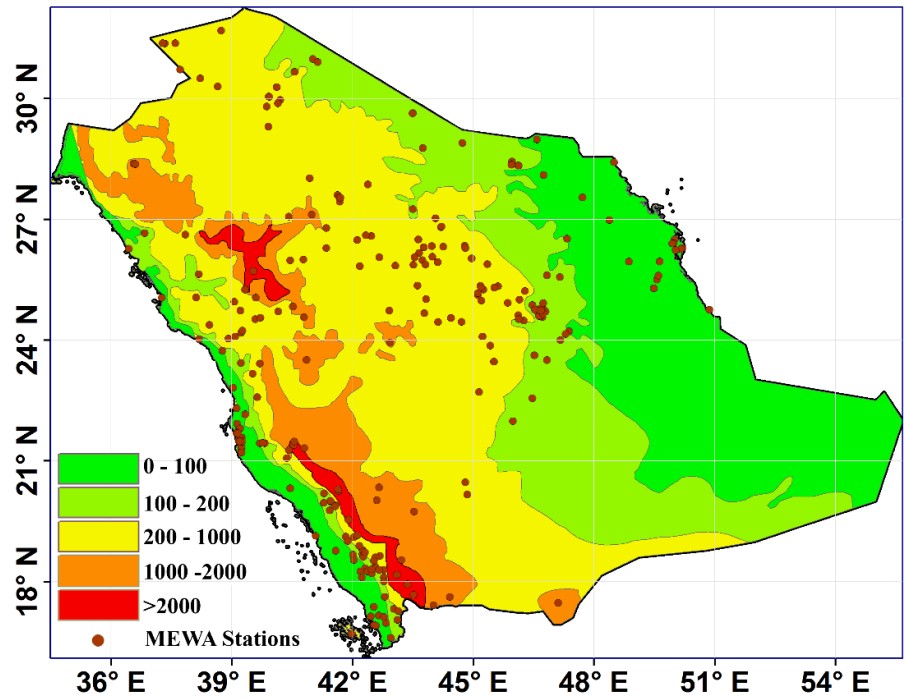

270

**Figure 3.** KSA rain gauges distribution over the different topographical regions.

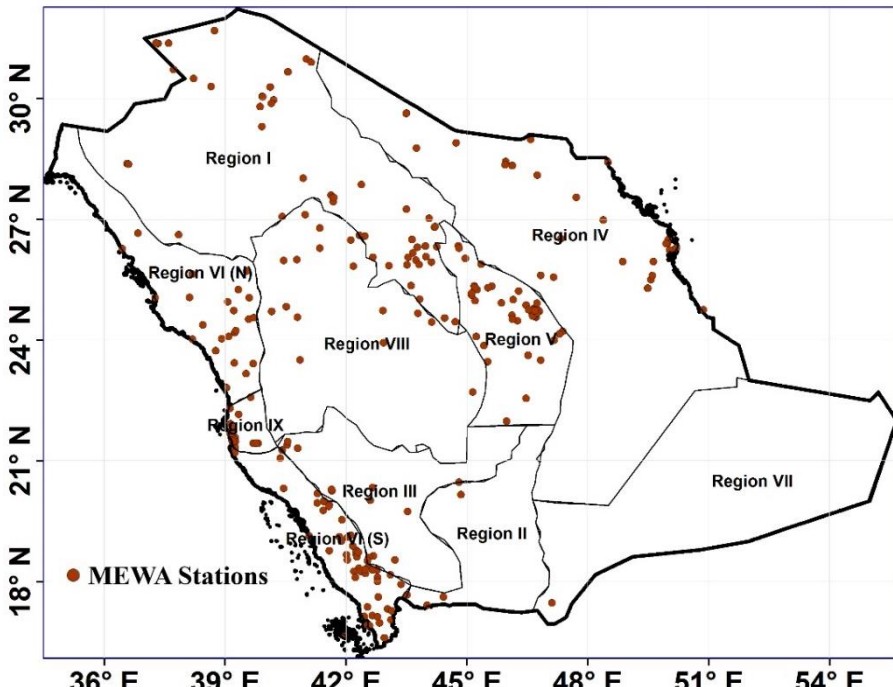

**Figure 4.** KSA rain gauge network distribution over the different hydrological regions.



## 5 Results

### 5.1 Seasonal-based Evaluation

Saudi Arabia has four seasons along the year; hence, the study period included 17 seasons in total (4 falls, 4 summers, 4 winters, and 5 springs). Table 3 illustrates the results of the performance indices of the three IMERG products, designed to guide the analysis of the performance of these products on a seasonal basis. The analysis was based on the three indices: categorical statistical indices, classical statistical indices, and correlation coefficient.

In terms of categorical statistical indices that indicate the detection accuracy, the three IMERG products performed irregularly during the four seasons over the study period. Most of the seasons showed high POD/CSI with an average of 0.9 while some seasons showed fair values decreasing to a minimum of 0.68. In general, IMERG-F and IMERG-L performed better than IMERG-E, and both had almost the same results during the study period. All IMERG products performed accurately in summer with high POD/CSI values ranging between 0.9 to 0.99. Spring comes after the summer in the level of detection accuracy where all IMERG products showed relatively high PODs/CSIs with an average of 0.897. Regarding fall seasons, the highest POD/CSI values, around 0.92, were observed in 2014 by the three IMERG products. The remaining years had lower POD/CSI values with an average of 0.854 with a slight improvement in IMERG-L and IMERG-F over IMERG-E. winter was the season with the least detection accuracy. IMERG-E showed only an acceptable PODs/CSIs ranging between 0.68 and 0.83. Whereas, both IMERG-L and IMERG-F showed an improvement in accuracy with higher PODs/CSIs values reaching 0.88.

Classical statistical indices (MAE, RMSE, and RBIAS) showed reasonably low error values compared to similar studies (Kim et al., 2017; Mahmoud et al., 2018, 2019) for IMERG products throughout the seasons. IMERG-F had the least MAE and RMSE values for all seasons (Table 3). Based on the calculated statistical errors, the seasonal analysis showed a clear improvement moving from IMERG-E to IMERG-L and finally to IMERG-F. In addition, all IMERG products showed very small values of RBIAS fluctuating between underestimation and overestimation of the rainfall during all the seasons (RBIAS < ± 0.7%), except for one calculation of RBIAS of IMERG-L on winter 2014-2015. The spring season exhibited the lowest estimated errors amongst other seasons for the three IMERG products, followed by the summer season. In the fall season, estimation errors for all the IMERG products showed similar trends; however, the averages of these errors were higher than those observed during spring and summer. The winter season had the highest errors (MAEs and RMSEs) compared to the rest of the seasons.

In term of consistency, IMERG products showed low CCs for all the seasons throughout the study period. However, the results revealed that IMERG-F had a better consistency than the other two products as it resulted in most of the highest CCs with a maximum CC (> 0.5 in fall 2017). Fall seasons showed the highest CCs compared to other seasons ranged between 0.185 to 0.557.



**Table 3. Statistical indices for the seasonal-based evaluation of IMERG products**

| Season | Year | CC IMERG-E | CC IMERG-L | CC IMERG-F | MAE IMERG-E | MAE IMERG-L | MAE IMERG-F | RMSE IMERG-E | RMSE IMERG-L | RMSE IMERG-F | RBIAS IMERG-E | RBIAS IMERG-L | RBIAS IMERG-F | POD/CSI IMERG-E | POD/CSI IMERG-L | POD/CSI IMERG-F |
|---|---|---|---|---|---|---|---|---|---|---|---|---|---|---|---|---|
| Fall | 2014 | 0.192 | 0.262 | 0.185 | 11.41 | 10.68 | 9.91 | 15.89 | 14.87 | 14.23 | 0.06 | 0.06 | -0.02 | 0.92 | 0.92 | 0.92 |
| | 2015 | 0.173 | 0.201 | 0.327 | 13.61 | 13.12 | 9.04 | 21.22 | 20.7 | 14.28 | 0.04 | 0.04 | -0.06 | 0.84 | 0.86 | 0.86 |
| | 2016 | 0.361 | 0.403 | 0.48 | 11.22 | 10.83 | 9.25 | 17.55 | 16.97 | 12.76 | -0.04 | -0.03 | -0.11 | 0.84 | 0.86 | 0.86 |
| | 2017 | 0.444 | 0.483 | 0.557 | 9.8 | 9.37 | 8.68 | 14.92 | 14.27 | 14.11 | -0.14 | -0.14 | -0.41 | 0.84 | 0.87 | 0.87 |
| Spring | 2014 | 0.113 | 0.118 | 0.342 | 7.7 | 7.81 | 6.45 | 11.59 | 11.58 | 10.37 | -0.13 | -0.12 | -0.36 | 0.9 | 0.91 | 0.9 |
| | 2015 | 0.21 | 0.089 | 0.216 | 10.89 | 11.45 | 8.24 | 16.76 | 18.04 | 13.06 | 0.02 | -0.01 | -0.17 | 0.91 | 0.95 | 0.95 |
| | 2016 | 0.273 | 0.222 | 0.415 | 10.47 | 9.98 | 9.42 | 16.95 | 16.13 | 16.15 | -0.04 | -0.04 | -0.07 | 0.91 | 0.93 | 0.93 |
| | 2017 | 0.289 | 0.096 | 0.251 | 9.69 | 8.75 | 8.16 | 14.89 | 13.41 | 12.72 | -0.02 | -0.13 | -0.12 | 0.89 | 0.92 | 0.88 |
| | 2018 | 0.033 | -0.248 | 0.228 | 12.85 | 9.19 | 6.83 | 18.31 | 12.96 | 10.34 | 0.15 | -0.66 | -0.1 | 0.88 | 0.73 | 0.89 |
| Summer | 2014 | 0.071 | 0.183 | 0.193 | 10.73 | 10.12 | 9.65 | 14.61 | 13.57 | 13.17 | -0.27 | -0.2 | -0.33 | 0.9 | 0.9 | 0.9 |
| | 2015 | 0.291 | 0.296 | 0.296 | 9.97 | 10.07 | 10.62 | 14.21 | 14.41 | 14.32 | -0.23 | -0.25 | -0.51 | 0.93 | 0.94 | 0.94 |
| | 2016 | 0.278 | 0.343 | 0.329 | 14.48 | 13.83 | 12.41 | 21.51 | 20.45 | 18.88 | -0.06 | -0.04 | -0.17 | 0.94 | 0.97 | 0.97 |
| | 2017 | -0.004 | -0.031 | -0.005 | 12.65 | 11.36 | 10.73 | 17.46 | 15.9 | 15.41 | -0.09 | -0.23 | -0.52 | 0.99 | 0.99 | 0.99 |
| Winter | 2014-15 | 0.082 | 0.101 | 0.094 | 11.97 | 11.58 | 10.47 | 15.15 | 14.78 | 10.39 | 0.31 | 5.68 | -0.5 | 0.78 | 0.71 | 0.8 |
| | 2015-16 | 0.095 | 0.119 | 0.147 | 12.09 | 12.58 | 11.36 | 16.32 | 16.5 | 12.28 | -0.05 | -0.03 | -0.1 | 0.72 | 0.73 | 0.74 |
| | 2016-17 | 0.106 | 0.194 | 0.356 | 15.63 | 13.25 | 12.19 | 21.66 | 20.56 | 20.24 | -0.2 | -0.19 | -0.18 | 0.68 | 0.76 | 0.76 |
| | 2017-18 | 0.301 | 0.317 | 0.286 | 14.23 | 14.28 | 12.89 | 19.35 | 21.35 | 14.22 | 0.43 | 0.55 | 0.16 | 0.83 | 0.87 | 0.88 |





On the other hand, it is worth to mention that the three IMERG products showed very weak correlations (CC < 0.5) during the seasons with slight enhancement in IMERG-F. This might be because Saudi Arabia is characterized by unreliable precipitation and high spatial and temporal temperature variability which makes the IMERG-F calibration done by GPCC, which is based on the monthly scale, not daily scale, sub-optimal (Tan and Santo, 2018). Therefore, it is better to use daily precipitation in the calibration as a further step to improve the IMERG-F algorithm. This is reinforced by similar findings and justifications

stipulated when the IMERG products were assessed over Malaysia by (Tan and Santo, 2018).

As for seasons, it can be noticed that based on the averages of the statistical measures, spring and summer were better represented by the IMERG products than fall and winter seasons. Summer is the drier season in Saudi Arabia that has the lowest amount of precipitation while the highest precipitation values occurred in spring (Hasanean and Almazroui, 2015). This indicates that IMERG products have the capability to detect seasons with the highest precipitation values as well as the seasons

with the lowest ones. On the other hand, IMERG products had relatively low performance in the winter; this may be due to the fact that most winter precipitation particularly falls over the southwest region which are mountainous areas (Hasanean and Almazroui, 2015). This will be corroborated by the topography-based evaluation in section 5.3.1.

## 5.2 Rainfall Intensity-based Evaluation

Overall, in terms of detection accuracy, the three products of IMERG presented high detection accuracy for the four various

rainfall classes over KSA over the period from March 2014 to June 2018. The observed PODs/CSIs ranged between 0.71 and 0.96. However, IMERG-F and IMERG-L showed higher performance in detecting all rainfall classes than IMERG-E. They both had the same detection accuracy with values of 0.96, 0.91, 0.9, 0.87, and 0.86 for storm events, heavy rainfall, moderate rainfall, light rainfall, and large storm events, respectively.

In terms of detection errors, it can be observed that the classical errors (MAE and RMSE) increased with the increase of rainfall

intensity for all IMERG products. Light rain, moderate rain, heavy rain, storm, and large storm events came in ascending order in term of errors. The values of error increased gradually with an average increase of about 10 mm except for storm events as it showed the highest errors with a significant increase (more than 50mm) compared to the class preceding it (storm). Regarding the RBIAS, very small percentages (0% to -0.69%) were observed for the three IMERG products during light, moderate and heavy rainfall. However, a clear underestimation was observed in large storm events amounting to around -11.83%, -12.64%

and -13.03% for IMERG-E, IMERG-L, and IMERG-F respectively. Overall, both light rain and moderate rain showed an acceptable level of errors, while the remaining rainfall classes had high errors compared to previous studies (Mahmoud et al., 2018). In general, IMERG-F exhibited the least errors in almost all the rainfall classes, while IMERG-E and IMERG-L fluctuated in performance across the different classes.

All the IMERG products had very low correlations with ground observations. This was observed for four classes of the rainfall,

with the exception of large storm events. Large storm events showed a higher correlation (0.74 observed by IMERG-L) in comparison to the other classes.



In summary, IMERG products showed high performance in capturing the various rainfall intensities. Light rain was the most detected class by IMERG products. Light rain accounted for about 60% of the total precipitation occurring over Saudi Arabia, which makes it particularly important to have a high detection accuracy for this class with minimum errors. The higher results

of CCs for large storms events may be due to the fact that these events are quite rare over Saudi Arabia; it represents less than 0.2% of the total precipitation. Thus, there is little probability of the temporal variability between the ground data and the satellite data for these large storms (i.e. higher CC). IMERG-F had the most accurate performance for most of the rainfall classes while it showed a close performance to IMERG-E and IMERG-L for the heavy rainfall and storm events as evidenced in errors (MAE and RMSE).

The multichannel GPM Microwave Imager (GMI) consists of 11 radiometric channels with a range of 10 GHz to 85.5 GHz, and modern Dual-frequency Precipitation Radar (DPR) that operates with both Ka-band (35.3 GHz) and Ku band (13.6 GHz) (Kim et al., 2017). These high-frequency GMI and DPR lead to the high detection accuracy of light rain by GPM products. Kim (Kim et al., 2017) assessed the performance of GPM (IMERG) products over Far-East Asia and they verified the high detection of light rain by IMERG products in comparison to TRMM (TMPA). Another study carried out by (Tan and Santo,

2018) to evaluate IMERG products against other satellite products over Malaysia revealed that IMERG products had the ability to detect light rain more accurately than other satellite products. Moreover, a study carried out by Xu et al. evaluated the IMERG and TRMM precipitation estimates over southern Tibetan Plateau region in China (Xu et al., 2017). Their results indicated that the IMERG products had a high capability to detect the light rain with an intensity of 0 to 5 mm/day. Moreover, we have noted that despite the small RBIAS in the various rainfall classes, except for the large storms, only the light rain

showed overestimations while the other classes showed underestimations. This finding is consistent with the findings of Tian (Tian et al., 2017). They also proved that light rain has lower PODs than heavy rain and that the POD increased with the rainfall intensity, which is also reflected in our results (Table 4).

### 5.3 Spatial Evaluation

The three GPM IMERG products were evaluated spatially by investigating the topographical effect and the performance of

the satellite over different hydrological regions. The following subsections will demonstrate the difference in the performance of the IMERG products in representing the spatial variation of rainfall.

### 5.3.1 Topographical evaluation

Fig. 5 shows the distribution of CCs and PODs over the five topographical regions of Saudi Arabia for the three IMERG products. From first glance, there was a clear enhancement in the performance of IMERG-F while IMERG-L had the lowest

performance. In terms of consistency, all IMERG products showed very low CCs ranging between 0.18 and 0.3. The coastal region, which has the lowest elevations (<100 m) and the inland region of Saudi Arabia that ranges between altitudes of more than 200 and less than 1000 showed similar trends of CC values. Lands that are adjacent to the coastal region with elevations less than 200 also showed an improvement in performance for IMERG-F as CC increased to 0.3 from less than 0.18 for both





IMERG-E and IMERG-L. The mountainous areas were divided into two regions: foothills and high mountains. Overall, all
the results for the different topographical regions presented improvements in the consistency observed in IMERG-F with the
ground observations except the foothills region.

**Table 4.** Rainfall intensity-based evaluation matrices.

| Event Date | Product | CC | MAE | RMSE | BIAS | POD/CSI |
|---|---|---|---|---|---|---|
| **Light** | IMERG-E | 0.093 | 7.51 | 13.04 | 0.03 | 0.84 |
| | IMERG-L | 0.075 | 8.68 | 17.17 | 0.04 | 0.87 |
| | IMERG-F | 0.145 | 4.26 | 6.66 | 0 | 0.87 |
| **Moderate** | IMERG-E | 0.045 | 12.59 | 16.49 | -0.01 | 0.89 |
| | IMERG-L | 0.039 | 13.32 | 18.83 | -0.01 | 0.9 |
| | IMERG-F | 0.073 | 11.41 | 13.39 | -0.03 | 0.9 |
| **Heavy** | IMERG-E | 0.012 | 23.58 | 26.55 | -0.1 | 0.9 |
| | IMERG-L | 0.034 | 23.18 | 26.36 | -0.09 | 0.91 |
| | IMERG-F | 0.113 | 24.46 | 26.55 | -0.14 | 0.91 |
| **Storm** | IMERG-E | 0.173 | 41.68 | 45.66 | -0.64 | 0.93 |
| | IMERG-L | 0.147 | 40.46 | 44.52 | -0.61 | 0.96 |
| | IMERG-F | 0.175 | 43.22 | 47.24 | -0.69 | 0.96 |
| **Large storm** | IMERG-E | 0.488 | 121.19 | 123.97 | -13.03 | 0.71 |
| | IMERG-L | 0.714 | 117.57 | 119.64 | -12.64 | 0.86 |
| | IMERG-F | 0.621 | 110.01 | 112.5 | -11.83 | 0.86 |

In terms of detection accuracy, the three IMERG products showed relatively high PODs ($> 0.85$), with some exceptions, which
indicates a high level of detection. IMERG-L and IMERG-F showed similar patterns and results of POD over the five
topographical regions, and they had slightly higher PODs than IMERG-E in three regions. The detection accuracy improved
in the foothills and high mountains regions for the IMERG-L and IMERG-F as well as the region adjacent to the coastal areas.
Fig. 6 shows the maps of the errors (MAE and RMSE) distribution for the three IMERG products over the various
topographical regions. It can be seen that there was a strong improvement in the results (fewer errors) for IMERG-F, while
IMERG-L had a lower performance than IMERG-E. Regarding MAE comparisons, the MAE of the inland, foothills and the
region adjacent to coastal areas remained the same in both IMERG-E and IMERG-L while it dropped to less than 5 mm in
IMERG-F. The estimated errors of the coastal region decreased to reach MAE between 7 and 10 mm for IMERG-F. The same
trend was observed for RMSE as the best product with the lowest RMSE in all topographical regions was IMERG-F, and the
worst one was IMERG-L. The most improvement was observed for the coastal and high mountains region for which RMSE
decreased from 10 mm to 15 mm for IMERG-E to 5 mm to 10 mm for IMERG-F. These regions represent the topographic
extremes, flat plain areas, and the highest altitudes, in the study area. Furthermore, the observed RMSEs in the inland region





and the region adjacent to the coastal areas decreased to less than 5 mm for IMERG-F. Minor percentages of RBIAS, almost negligible, were observed for all the IMERG products as seen in Table 5.

The results indicate the high detection accuracy of IMERG-F in different topographical regions. Even though all the IMERG
products presented low correlation values with ground observations over the different topographical regions, IMERG-F showed a good improvement compared to IMERG-E and IMERG-L. Moreover, the coastal, foothills and high mountains regions exhibited the highest errors compared to other topographical regions.

This conclusion is in agreement with the findings of a previous study conducted by Prakash (Prakash et al., 2018) who assessed the performance of IMERG products over India. They claimed that IMERG products were affected by the orographic process,
which leads to higher errors and negative bias in mountainous areas. Another study carried out by Kim (Kim et al., 2017) also revealed the drawbacks of IMERG products over the mountainous and coastal regions. They attributed the poor performance of IMERG at coastal regions to a deficiency in the calibration algorithm that identifies rainy clouds over coastal areas. In the same context, similar results obtained by Anjum (Anjum et al., 2018) prompted them to caution users from using IMERG products in mountainous areas because of the high uncertainty in the daily precipitation, particularly light rainfall.

**Table 5.** RBIAS measurements for IMERG-E, IMERG-L, and IMERG-F using the topographical-based evaluation.

| Region Name | Altitude (m) | IMERG-E | IMERG-L | IMERG-F |
|---|---|---|---|---|
| Coastal region | 0-100 | 0 | -0.03 | 0.01 |
| Areas adjacent to the coasts | 100 to 200 | -0.02 | -0.03 | -0.01 |
| Inland region | 200 to 1000 | 0.01 | -0.02 | 0.02 |
| Foothills region | 1000 to 2000 | -0.08 | -0.25 | 0.11 |
| High mountains region | More than 2000 | -0.09 | -0.23 | -0.05 |





**Figure 5.** Evaluation of IMERG products for the different topographical regions using CC measurements for a) IMERG-E, b) IMERG-L, and c) IMERG-F and POD indicator for d) IMERG-E, e) IMERG-L, and f) IMERG-F.






**Figure 6.** Evaluation of IMERG products for the different topographical regions using MAE measurements for a) IMERG-E,
b) IMERG-L, and c) IMERG-F and RMSE indicator for d) IMERG-E, e) IMERG-L, and f) IMERG-F



### 5.3.2 Hydrological Regions Assessment

Fig. 7 demonstrates the results of two groups of performance measures (CC and POD/CSI) across the different hydrological
regions over Saudi Arabia. Region VII that is located in the southeast of the country and region II which is located in the south
of the country had almost perfect correlations (CC = 1) while low to moderate correlation (CC is 0.3 to 0.5) was observed in
one region (region V) that is located in the middle of the country. The rest of the regions showed low correlations (CC < 0.5).
Correlations visibly improved for IMERG-F to reach CC > 0.5 throughout most of the regions.

The three IMERG products showed a very high probability of detection in all regions except region VII, which presented lower
POD for the three IMERG products (0.75). It is observed (Fig. 7) that IMERG-L and IMERG-F showed the same high detection
accuracy with superiority for IMERG-L.

Fig. 8 presents the classical errors (MAE, RMSE) across the hydrological regions. The performance of the three products can
be ranked as IMERG-F, IMERG-L, and IMERG-E in descending order. Most of the regions showed lower MAEs for IMERG-
F. The MAEs for IMERG-F were about 2 mm lower than MAEs for IMERG-L in each region. The MAEs observed in region
II, region VI south and region VII did not show any improvement through the IMERG (-E-L-F) products, nonetheless region
II already had the lowest MAEs (<5 mm). The same trend for the order of the three IMERG products was observed in the
calculated RMSEs. The minimum RMSE was observed in three regions while the maximum RMSE occurred in two regions.
Regarding RBIAS, it is shown in Table 6 that IMERG products had insignificant fluctuation of RBIAS percentages for most
of the hydrological regions. However, relatively high underestimation reflected by RBIAS percentages were observed in two
regions (region II and region VII) for the three IMERG products with an average of -10.813% and 17.137% for region II and
region VII respectively. In addition, region II had the lowest MAE and RMSE while it had the highest RBIAS percentages for
the three IMERG products.

In conclusion, IMERG-F outperformed IMERG-E and IMERG-L in the hydrological-based evaluation in terms of consistency
and errors. Region VII and region II showed perfect correlation with the ground observations this probably resulted from the
fact that both regions contain few rain gauge stations (one and two respectively) which leads to very high correlation. Region
VII showed the lowest POD values compared to other regions. Overall, low density of rain gauge stations makes it inefficient
to evaluate satellite products and results in an underestimation of the performance of these products. This is in agreement with
the results of a previous study that investigated the relationship between the density of rainfall gauge stations and the
performance of IMERG products (Tian et al., 2017).

**Table 6.** RBIAS measurements for IMERG-E, IMERG-L, and IMERG-F using the hydrological-based evaluation.

| Regions | IMERG-E | IMERG-L | IMERG-F |
|---|---|---|---|
| Region I | 0.08 | 0.11 | -0.02 |
| Region II | -10.93 | -5.04 | -16.47 |
| Region III | -0.08 | -0.06 | -0.24 |
| Region IV | 0.04 | 0.07 | -0.02 |



| Region V | 0.02 | 0.04 | -0.02 |
| Region VI-North | 0.02 | 0.13 | -0.27 |
| Region VI-South | -0.02 | -0.01 | -0.03 |
| Region VII | -16.49 | -17.33 | -17.59 |
| Region VIII | 0.1 | 0.32 | -0.13 |
| Region IX | -0.13 | -0.23 | -0.7 |







**Figure 7.** Evaluation of IMERG products for the different hydrological regions using CC measurements for a) IMERG-E, b) IMERG-L, and c) IMERG-F and POD indicator for d) IMERG-E, e) IMERG-L, and f) IMERG-F







**Figure 8.** Evaluation of IMERG products for the different hydrological regions using MAE measurements for a) IMERG-E, b) IMERG-L, and c) IMERG-F and RMSE indicator for d) IMERG-E, e) IMERG-L, and f) IMERG-F






# 6 Conclusions

This study assessed the performance the GPM three IMERG products; early, late, and final-run product, over KSA. Ground
observations from 275 rain gauge stations over the period March 2014 to June 2018 were used as a reference. The evaluation
of IMERG satellite products was carried out, considering three approaches. The first approach evaluated the seasonal effect
on the performance of IMERG products; the second was to evaluate the impact of spatial characteristics, represented by the
topographical and hydrological zones, on the accuracy of IMERG products; and the last approach focused on evaluating the
performance of IMERG in detecting rainfall at different intensities. Quantitative statistical indices were used to quantify the
performance of the IMERG products. The main conclusions of this study can be summarized as follows:

- The seasonal analysis showed an improvement in the performance from IMERG-E, to IMERG-L, to IMERG-F. Nevertheless, all IMERG products showed very weak correlations (CC < 0.5) with ground observations throughout all the seasons.

- Spring and summer are the most detected seasons by IMERG products. This leads to a conclusion that IMERG
products have the capability to detect seasonal rainfall with both the highest (maximum daily rainfall observed on spring) and the lowest precipitation.

- It was interesting to observe the high performance of IMERG products across the various rainfall intensity classes. According to the calculated classical statistical indices, the light rain had the lowest detection errors by IMERG products. However, the higher the rainfall intensity, the higher the detection errors in the IMERG products. The
detectability of the rainfall, as indicated by POD, exhibit a reversed trend with moderate, heavy rainfall and storms having gradually higher detectability.

- Even though the CC values are generally low for different rainfall intensities, large storm events showed significantly higher CCs (0.5 to 0.7) compared to lower intensity events. This is probably induced by the rarity of such large storms over arid regions such as Saudi Arabia.

- Results of the evaluation based on hydrological regions were similar to other evaluations as IMERG-F outperformed IMERG-E and IMERG-L in terms of consistency and estimated errors. Hydrologic Region VII and region II showed perfect correlation with ground observations. This probably resulted from the fact that both regions contain few rain gauge stations (one and two respectively) which could have led to such a high correlation. Contrarily, region VII showed the lowest POD values compared to other regions. Overall, low density of rain gauge stations hinders the
proper evaluation of satellite products and tends to underestimate the performance of the products.

- Topographical features had a significant influence on the performance of IMERG products. The detectability (POD) was improved significantly in higher altitudes (mountains, and foothills regions), particularly for IMERG-F. However, the areas adjacent to the coasts showed a significant reduction in the estimation errors of IMERG-F; whereas the highest estimation errors were observed in coastal regions, foothills, and mountainous regions.





*Acknowledgement* This work was supported by the National Water Center at United Arab Emirates University (UAEU) under grants no. 31R150 and 31R191.

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
