# Peer review of "Performance of GPM-IMERG precipitation products under diverse topographical features and multiple-intensity rainfall in an arid region"

_Hydrology and Earth System Sciences, 2019_

## Short Comment (SC1) · 10 Feb 2020

Please identify the IMERG version used (e.g., V05B, V06B). This information should be included in the relevant data section, and would be helpful if it is also included in the abstract. Please cite the relevant documentation appropriate for the version of IMERG used; for V06, the reference is:

Huffman, G. J., Bolvin, D. T., Braithwaite, D., Hsu, K., Joyce, R., Kidd, C., Nelkin, E. J., Sorooshian, S., Tan, J., Xie, P., 2019: Algorithm Theoretical Basis Document (ATBD) Version 06. NASA Global Precipitation Measurement (GPM) Integrated Multi-satellitE Retrievals for GPM (IMERG). NASA, https://pmm.nasa.gov/data-access/downloads/gpm.

---

## Short Comment (SC2) · 13 Feb 2020

This work focused on the impact of several crucial factors including elevation, season, hydrological region and precipitation intensity on the precipitation retrievals of IMERG suite (i.e., Early, Late and Final runs) over Saudi Arabia. This is a very lengthy study, covering several important factors that influence the performance of satellite precipitation retrievals. However, I noted some issues that need clarification.

a. The pertinent studies investigating the impact of these four crucial factors on the

errors of SPPs are neglected in Introduction.

b. The objective of this study is clear. These several factors, including topography, season, hydrological region and precipitation intensity are crucial factors that affect the retrieval accuracy of satellite precipitation products (SPPs). However, several studies have investigated the impact of these four factors on the precipitation retrievals of SPPs but are neglected by authors. For example, Chen et al. (2019) evaluated the performance of SPP, separating the analysis by the input sources (IR or MW sensors). Using a set of evaluation metrics, they examined how these input sources (i.e., IR and PMW) performed overall, by climate region, by elevation, and by season. Thus, the major scientific questions between this article and above article are similar.

c. Several studies have confirmed that the errors of SPPs are related to the rainfall intensity, such as Kirstetter et al., 2013, Chen et al., 2013, Chen et al., 2020. Chen et al. (2020) even found that a power function is observed between the retrieval accuracy and the precipitation intensity, indicating that the errors of SPPs increase with precipitation intensity. Thus, the differences (including but not limited to findings, methodology) between this article and mention-above articles should be given or discussed.

Therefore, I suggest the authors further supply a comprehensive research background for this work and completely clarify the differences between this paper and similar papers.

I hope that the comments will help the authors to further improve this manuscript.

References

Chen, H., Yong, B., Gourly, J. J., Liu, J., Ren, L., Wang, W., Hong, Y., Zhang, J., 2019. Impact of the Crucial Geographical and Climatic Factors on the Input Source Errors of GPM-based Global Satellite Precipitation Estimates. J. Hydrol. 575: 1-16, https://doi.org/10.1016/j.jhydrol.2019.05.020.

Chen, H., Yong, B., Shen, Y., Liu, J., Hong, Y., Zhang, J., 2020. Comparison analysis

of six purely satellite-derived global precipitation estimates. J. Hydrol. 581, 124376, https://doi.org/10.1016/j.jhydrol.2019.124376.

Chen, S., Hong, Y., Cao, Q., Gourley, J. J., Kirstetter, P. E., Yong, B., Tian, Y., Zhang, Z. X., Shen, Y., Hu, J. J., Hardy, J., 2013. Similarity and difference of the two successive V6 and V7 TRMM multisatellite precipitation analysis performance over China. J. Geophys. Res.-Atmos. 118(23).

Kirstetter, P. E., Hong, Y., Gourley, J. J., Schwaller, M., Petersen, W., Zhang, J., 2013. Comparison of TRMM 2A25 products, version 6 and version 7, with NOAA/NSSL ground radar–based National Mosaic QPE. J. Hydrometeorol. 14(2), 661-669.

---

## Author Comment (AC1) · 18 Feb 2020

Thanks for the valuable comment. The IMERG version will be added to the final manuscript after receiving the reviewers' comments. A respective reference will be cited as per the version of IMERG used.

---

## Referee Comment (RC1) · Husain Najafi (Referee) · 21 Feb 2020

Three different products of Global Precipitation Measurement - Integrated Multi-satellitE Retrievals from Global Precipitation Measurement (GPM-IMERG) are compared to a reference rainfall dataset in the Kingdom of Saudi Arabia. Complex topography, and rainfall intensity events are introduced as two challenges of satellite precipitation products addressed in the paper. GPM-IMERG is evaluated over the period between March 2014 to June 2018 for five topographical classes, ten hydrological regions and five rainfall intensity classes. Six metrics based on the contingency table

and continuous evaluation were calculated and reported for four seasons (fall, spring, summer and winter).

General Comments

Data and methodological approach of any research study must lead to robust scientific findings. Otherwise, the results will be doubtful for scientific community. Research findings based on reliable data and robust methods will support further improvement of available satellite precipitation products. The scientific methods provided in the current study are valid; however, some specific aspects have not been outlined clearly. The novelty cannot be considered as substantial. Scientific significance is limited in provision of new concepts, tools, and data. The performance of GPM-IMERG products was evaluated in different parts of Arabian Peninsula recently in two research studies listed in references (Mahmoud et al., 2018; 2019) using similar performance measures. Therefore, the novelty of submitted manuscript required justification.

Specific comments

(1) Research questions are not clearly mentioned in the introduction.

(2) In the first section of the paper, some aspects of advances and challenges of evaluating satellite precipitation products is discussed. Extensive introduction is provided on precipitation measurements with specific focus on remote sensing of precipitation. However, literature on the result of other studies which evaluate satellite-based products in arid regions is not sufficiently elaborated. Findings from other research studies which assessed the performance of GPM-IMERG products in regional scale in general, and the area under study in specific, is not provided.

(3) Seasonal and annual rainfall based on Ministry of Environment, Water, and Agriculture (MEWA) rain gauges are not provided in section 2. This information can provide a basis for comparison between reference data and GPM-IMERG.

(4) Since the research is based on local rainfall data, access protocol to the reference

data is not provided. It is mentioned in line 194 that the data was downloaded from MEWA website; however, the corresponding link is not provided. Relevant aspects of reproducibility of scientific result is not addressed.

(5) The overall quality of selected reference data (MEWA stations) is not mentioned in section 3.1. Quality control procedure applied to MEWA data is not explained. Previous studies which have used this dataset are not cited. Convincible reasons is required on why MEWA is selected as the reference data of the research besides the reason mentioned in lines 172-174. It was helpful if mentioned which authorities are responsible for recording rainfall within the study area, and to provide some arguments on reliability of MEWA compared to other rainfall sources. The type of MEWA rain gauges is not mentioned.

(6) Domain selection requires careful attentions for evaluating a satellite precipitation product in the absence of a dense rain gauge network. Although two important aspects (topographical effect and evaluating satellite products in hydrological regions) have been studied to investigate the performance of GPM-IMERG, the results are likely to be sensitive and dependent to some unknown extent on the spatial evaluation described in section 4.3.3. The low density and spatially non-uniform rainfall network selected as reference in this research will influence spatial evaluation of GPM-IMERG in at least some regions reported in section 5.3. There is not any rain gauge station located between 16 °-24 °N and 48 °-55 °E. Given the low density of MEWA stations in regions I, II, III, IV, VII, VIIII, and their corresponding topographical classes, robustness of the results provided in section 5.3.1 and 5.3.2 is under question and not straightforward. It is explicitly declared in the conclusion section of the manuscript (lines 466-469) that the issue (low density of rain gauge stations) prevents a proper evaluation of the rainfall satellite product. Results provided in table 6 for hydrologic regions number II, III and VII also provide evidence that the highest percentage of relative bias (RBIAS) are calculated in those areas incorporating small number of gauge stations (see Figure 4). This argument is critical and requires careful considerations as it could highly effect the

result.

(7) In lines 136-138, it is stated that a comprehensive evaluation is presented in the current study. However, often-used metrics namely statistical distribution and metrics on extended contingency table have not been considered.

(8) It is not apparent how the methodology described in line 206 as "point to point analysis" is used for spatial evaluation (explained in section 4.3.3). The research method used in generating figures 5 and 6 is not clear. How metrics provided in Table 1 are calculated for spatial evaluation? How Probability of Detection (POD) is calculated over the five topographical classes, and ten hydrological regions? Relevant formulation on pointwise analysis and areal-average evaluation is not provided for metrics in Table 1.

(9) Figure 3 and Figure 4 provide the spatial coverage of MEWA rain gauges within correspondent topographical and hydrological regions. However, the percentage of areas correspond to each class and corresponding percentage of MEWA stations is not provided. The number of rainfall events for each intensity class is not provided.

(10) Evaluation methods for comparing satellite rainfall to gauge-based products have limitations which are not addressed.

(11) In section 4.1, it is mentioned that rainfall events are determined during March 2014 to June 2018 based on ground observations. However, missing rate is not provided. The way missing data is treated to detect rainfall events is not mentioned in the research methodology.

(12) The sentence "The main superiority of satellite data over rain gauge data is that it provides uniform spatial coverage at high temporal resolution" stated in lines 53 and 54 is subject to argument.

(13) Lines 97-102: It is argued that availability of rain gauges in mountains areas is not common. The statement is general, and might not hold valid for some regions.

(14) Same as above, Lines 167-168: The sentence "Many researchers used the GPCC

(http://www.dwd.de/) gauge data for evaluation purposes; this data is not of sufficient density nor distribution for obtaining ground observation data at fine spatial resolution (Wang et al., 2017)" is a general statement and is subjected to arguments.

(15) Lines 243-247: How seasonal evaluation of GPM-IMERG products during March 2014-June 2018 could help to bring a better understanding of the climate of Saudi Arabia and monitoring climate change in the region?

Technical comments

(16) Sentence in line 128 requires revision.

(17) Geographical coordination of area under study provided in lines 142 and 143 requires revision.

(18) Figure 1 does not have a legend. Both MEWA stations and major cities within the study area are represented in black dots.

References:

(I) Mahmoud, M T., Al-Zahrani, M A. and Sharif, H O. 2018. Assessment of global precipitation measurement satellite products over Saudi Arabia, J. Hydrol., 559: 1–12. doi:10.1016/j.jhydrol.2018.02.015.

(II) Mahmoud, M T., Hamouda, M A. and Mohamed, M M. 2019. Spatiotemporal evaluation of the GPM satellite precipitation products over the United Arab Emirates, Atmos. Res., 219: 200–212. doi:10.1016/j.atmosres.2018.12.029.
* * *

---

## Author Comment (AC2) · 4 Mar 2020

Thanks for the valuable comment. It is always a struggle for an author when trying to provide a comprehensive background at the risk of the article being mistaken for a "review article" rather than a "research article". The authors will try to include the suggested pertinent studies where they best fit in the introduction and discussion sections. Particularly Chen et al. (2019) as it is directly related to the investigation carried out in this study.

[Figure]

References Chen, H., Yong, B., Gourly, J. J., Liu, J., Ren, L., Wang, W., Hong, Y., Zhang, J., 2019. Impact of the Crucial Geographical and Climatic Factors on the Input Source Errors of GPM-based Global Satellite Precipitation Estimates. J. Hydrol. 575: 1-16, https://doi.org/10.1016/j.jhydrol.2019.05.020.

---

## Referee Comment (RC2) · Anonymous Referee #2 · 26 Mar 2020

This paper presents quite an exhaustive summary of a data comparison of gauge- and satellite-based measurements during 4 years for the semi-arid region of Saudi Arabia. The main focus lies on the performance of the satellite product (GPM-IMERG) with respect to topography, rainfall intensity, and season. Main results include that the first half of the year shows better performance than the second, smaller rainfall better than larger, and coastal areas and mountains worse than inland.

With all its exhaustiveness in reporting the numbers, the study reads like a technical report for a research project rather than a scientific article. With respect to the science,

not many of the most obvious questions were answered or raised at all, as detailed below. Because I do not see an easy way to transform this report into a scientific study, I must reject it for further publication in HESS.

Like the other reviewer noted, there is a strong overlap (topic- and author-wise) with two recent studies (https://doi.org/10.1016/j.atmosres.2018.12.029 and https://doi.org/10.1016/j.jhydrol.2018.02.015), who conduct the same statistical comparison of the same satellite data in the same or a neighboring region. In response, the authors argue that more years have been studied now, or hydrological zones have been considered instead of political ones. It is true that more years are covered now, but each year is still treated separately, so there is no real gain in significance. More generally, it is not clear what scientific differences there are between here and there, so the overlap remains.

As for the science, my impression is that if the authors had started, and they should have, from a map of seasonal rainfall climatology for the area, it would explain much of the presented results. Related to that, many claims are not really surprising, such as that satellite sensors fail to detect smaller showers or that rainfall errors scale with rainfall magnitude. If the authors decide to try to put this into a scientific article, I recommend strong reduction (why are there so many statistical measures?) and focus on the essential results, include climatology information and argue from that. It is likewise important that each result is augmented by a solid significance analysis, given that only 4 years of data have been used and external factors play an important role. This is especially important since not much aggregation has been undertaken with the data, leading to so many single questionable results instead of a few with greater significance.

More details:

l 15: Why three?

l 18: Isn't that the purpose of doing the final run? – If that doesn't improve results it

would not be done.

Abstract: The text should focus more on the surprises of the analysis. As it is written, the results are exactly as one would expect.

l 52: "sub-par"?

Intro: Most of the Introduction is about (already known) satellite technology and should be removed.

l 126: "for hydrological..."?

l 140: This chapter could be significantly shortened. Giving the key characteristics is enough.

l 69: "...nor distribution"?

l 97ff: again too technical and not of interest.

l 205: "grids points"

Figure 2: If the Figure describes what is in the preceding text it should be removed.

l 205: You should discuss whether and why there are no false positives (IMERG>0, MEWA=0) in the IMERG data.

l 221: That is a false characterization of the CSI. Please correct, and please explain why you choose more than one index.

l 222: If they give the same values then only one should be used.

l 223: This should be moved to l 205.

l 225: This is again a false characterization of MAE and RMSE. Both are closely related, and it should be justified clearly if both are reported.

Table 1: HESS readers should know about this, so it can be moved to an appendix or cited from the literature. Or one can simply cite the Mahnoud et al. (2019) study.

l 242: No reference is necessary for the definition of seasons.

l 243ff: This should be removed or merged with the introduction.

Figure 3, legend: please label altitude

l 276: The result should be presented for each season, not for each season and year. And if you decide otherwise, the reporting should at least mention the typical variation between years and try to understand that.

l 281: irregularly because you have such small samples, see previous comment.

l 289: First you show a large table, and then you report almost every number in it.

l 325: The fact that rainfall errors increase with rainfall magnitude is a normal scaling behavior. I would be surprised if it were otherwise.

l 345: Remove or move to Introduction.

Fig. 5: This looks like single figure with 6 different scalings. Comparing it to Fig. 3 we obviously see topography here. Moreover, it gives only little information because the main features are probably below significance anyway. Why is the CC pattern opposite between IMERGE-L and IMERGE-F?

Fig. 7: Is the difference to Fig. 5 only that for Fig. 7 all stations are aggregated for each region?

---

## Author Comment (AC4) · 2 Apr 2020

The authors would like to thank Reviewer#2 for the time invested in reviewing our manuscript. Some of the reviewers' comments offer valuable insights for revising and improving this manuscript. We have followed the comments carefully and responded to each comment or suggestion.

This paper presents quite an exhaustive summary of a data comparison of gauge- and satellite-based measurements during 4 years for the semi-arid region of Saudi

Arabia. The main focus lies on the performance of the satellite product (GPM-IMERG) with respect to topography, rainfall intensity, and season. Main results include that the first half of the year shows better performance than the second, smaller rainfall better than larger, and coastal areas and mountains worse than inland. With all its exhaustiveness in reporting the numbers, the study reads like a technical report for a research project rather than a scientific article. With respect to the science, not many of the most obvious questions were answered or raised at all, as detailed below. Because I do not see an easy way to transform this report into a scientific study, I must reject it for further publication in HESS.

Reply: The authors disagree with the reviewer with regards to what counts as a contribution to science. Novelty and contribution to science can take several forms. This is definitely not the first, nor is it going to be the last, article published on the evaluation GPM IMERG products under different topographical, climatic, and hydrological conditions. As such the authors did their best to apply state-of-the art methods in evaluating GPM satellite products over one of the least represented regions in literature, the Arabian Peninsula. Also, the authors disagree that this article reads as a technical report since they have had extensive experience writing scientific articles and can easily make such a distinction. We would have appreciated a more constructive review pointing out the "obvious questions" that we may have missed, and we would have taken on the necessary effort to include those in the revised article, but the reviewer chose to deny us this chance.

Like the other reviewer noted, there is a strong overlap (topic- and author-wise) with two recent studies (https://doi.org/10.1016/j.atmosres.2018.12.029 and https://doi.org/10.1016/j.jhydrol.2018.02.015), who conduct the same statistical comparison of the same satellite data in the same or a neighboring region. In response, the authors argue that more years have been studied now, or hydrological zones have been considered instead of political ones. It is true that more years are covered now, but each year is still treated separately, so there is no real gain in significance. More

generally, it is not clear what scientific differences there are between here and there, so the overlap remains.

Reply: The authors do not see any problem with their attempt to conduct state-of-the-art evaluation methods to a study area that does not appear often in literature. The reviewer's claim that the overlap of authors is grounds for rejection is not founded, many research groups publish several articles on the same study area. If anything, this shows that our research group is keen on improving and expanding the scientific methods applied to our study region. We were forthcoming in noting the differences between this article and the previously published ones. We would like to note that the methods applied have been subject to improvement and we will continue to improve our methods and obtain better data to enhance our evaluation, so you can expect to see more articles in the future that covers the same study area. In addition, neither the techniques used, nor the scientific outcomes match the articles mentioned by the reviewer. In this article we have focused on two new outcomes (1) evaluating the effect of the topographical features of the study area on the satellite detection accuracy, and (2) evaluating the capability of the satellite products in detecting light rain. In the last point raised the reviewer argues that the use of data from multiple years does not represent an improvement in the evaluation of GPM IMERG data. Perhaps the reviewer is not familiar with the scarcity of ground observation data in the region (Arabian Peninsula), where obtaining multi-year country-wide data is in itself an achievement. The authors did not only treat the data obtained from each year separately, but we also provided an overall discussion of the performance in the 4 years of data (Lines 275-304). If we are given a chance to improve our manuscript presentation we would add rows to Table 3 to present a collective performance evaluation of GPM IMERG products throughout the 4 years.

As for the science, my impression is that if the authors had started, and they should have, from a map of seasonal rainfall climatology for the area, it would explain much of the presented results. Related to that, many claims are not really surprising, such

as that satellite sensors fail to detect smaller showers or that rainfall errors scale with rainfall magnitude. If the authors decide to try to put this into a scientific article, I recommend strong reduction (why are there so many statistical measures?) and focus on the essential results, include climatology information and argue from that. It is likewise important that each result is augmented by a solid significance analysis, given that only 4 years of data have been used and external factors play an important role. This is especially important since not much aggregation has been undertaken with the data, leading to so many single questionable results instead of a few with greater significance.

Reply: The authors would like to thank the reviewer for his valuable suggestions. The seasonal rainfall variation was requested by the first reviewer and was included in our authors' comment AC3 (Mohammed et al. 2020). Nevertheless, the requested map can be produced and included in the final revision of the manuscript. Moreover, the authors believe that the implemented statistical measures complement each other and are very important for the full evaluation of GPM IMERG products. Mainly these statistical measures are used to assess three main issues: detection accuracy, errors, and the consistency (Lines 215-229). These measures were adopted by almost all previous research in this field.

More details: L 15: Why three?

Reply: because they are the main products produced by the GPM satellite and are subject to investigation by many research articles.

L 18: Isn't that the purpose of doing the final run? – If that doesn't improve results it would not be done.

Reply: This is not always the case, had the reviewer checked other recent articles (including our latest publication Mahmoud et al. 2019) it would have caught his/her attention that the Final product does not necessarily have the best performance.

Abstract: The text should focus more on the surprises of the analysis. As it is written, the results are exactly as one would expect.

Reply: The authors disagree with this statement entirely. Scientific investigations report on both expected and unexpected results.

L 52: "sub-par"?

Reply: If the reviewer is proposing that we use "subpar" without the hyphen, then we can definitely do that in the revised manuscript. Otherwise a synonym can be used.

Intro: Most of the Introduction is about (already known) satellite technology and should be removed.

Reply: We appreciate the reviewers' comment, it would have been more helpful if the reviewer had specified the sections that are deemed unnecessary.

L 126: "for hydrological..."?

Reply: The sentence will be modified to read "TRMM cannot be the only source of data for hydrological applications as it provides limited input information"

L 140: This chapter could be significantly shortened. Giving the key characteristics is enough.

Reply: Thank you for your comment. This section will be improved in light of the reviewer's comments in the revised manuscript.

L 169: "...nor distribution"?

Reply: The distribution of the stations used to produce the GPCC data is not representative for the study area because it relies on few stations.

L 197: again too technical and not of interest.

Reply: Thank you for your comment. However, we have received interest from previous reviewers, on other related publications, to include such technical information to

improve the replicability of the results.

L 205: "grids points"

Reply: The phrase will be corrected to "grid points".

Figure 2: If the Figure describes what is in the preceding text it should be removed.

Reply: Thank you for your comment. The authors will reduce the text and keep the figure for better illustration and understanding of the module that we used. This is thought to improve the replicability of results.

L 205: You should discuss whether and why there are no false positives (IMERG>0, MEWA=0) in the IMERG data.

Reply: The authors have discussed this particular point in Lines 221 to 223. Since, the analysis focused on matching only datapoints where rain was observed by gauge stations with that of satellite data, it did not include false positives. However, we will try to add a few lines discussing the false positives in the dataset.

L 221: That is a false characterization of the CSI. Please correct, and please explain why you choose more than one index. l 222: If they give the same values then only one should be used.

Reply: The authors agree with the reviewer and intend to remove the statement and calculation of CSI as it does provide additional information about the satellite performance in this study.

L 223: This should be moved to l 205.

Reply: Noted.

L 225: This is again a false characterization of MAE and RMSE. Both are closely related, and it should be justified clearly if both are reported.

Reply: The authors will correct this statement in the revised manuscript to read "The

MAE and the RMSE are indices that, when used together, can assess the variation in the errors of a dataset. The RMSE is either larger than or equal to the MAE. This difference between RMSE and MAE correlates with the variance in the individual errors in an analyzed dataset."

Table 1: HESS readers should know about this, so it can be moved to an appendix or cited from the literature. Or one can simply cite the Mahmoud et al. (2019) study.

Reply: The authors believe that including the equations used is important as a reference for the reader. Should the editor suggest to move Table 1 to the appendix, we would not object to that.

L 242: No reference is necessary for the definition of seasons.

Reply: Thank you for your comment. We will remove the citation.

L 243: This should be removed or merged with the introduction.

Reply: Thank you for your comment. We will remove this part.

Figure 3, legend: please label altitude

Reply: The authors already made the corrections in their response to the first reviewer AC3 (Mohammed et al. 2020)

L 276: The result should be presented for each season, not for each season and year. And if you decide otherwise, the reporting should at least mention the typical variation between years and try to understand that.

Reply: The authors appreciate the reviewer's valuable comment. The authors tried to give such overall discussion in the text. However, we will try to include give more focus on the overall seasonal accuracy rather than including the differences of between the years.

L 281: irregularly because you have such small samples, see previous comment.

Reply: Noted, the authors will revise this discussion to avoid any generalization.

L 289: First you show a large table, and then you report almost every number in it.

Reply: The authors will try to reduce repetition without compromising the discussion.

L 325: The fact that rainfall errors increase with rainfall magnitude is a normal scaling behavior. I would be surprised if it were otherwise.

Reply: Nothing is a "fact" unless proven by reliable investigations. As far as the authors know this is one of the few attempts to evaluate GPM IMERG performance in detecting light rain.

L 345: Remove or move to Introduction.

Reply: Noted. We will move this part in the introduction.

Fig. 5: This looks like single figure with 6 different scalings. Comparing it to Fig. 3 we obviously see topography here. Moreover, it gives only little information because the main features are probably below significance anyway. Why is the CC pattern opposite between IMERGE-L and IMERGE-F?

Reply: Figure 5 is not a single figure with 6 different scalings, but rather 6 small figures of the results of the CC and POD for the three satellite products IMERG-E, L, and F.

Fig. 7: Is the difference to Fig. 5 only that for Fig. 7 all stations are aggregated for each region?

Reply: The aggregation for each figure depends on the feature we intend to analyze. Figure 5 is needed to evaluate the impact of topography and Figure 7 illustrates the variation in the performance of satellite data over the different hydrological regions.

References Mahmoud, M. T., Hamouda, M. A. and Mohamed, M. M.: Spatiotemporal evaluation of the GPM satellite precipitation products over the United Arab Emirates, Atmos. Res., 219(January), 200–212, doi:10.1016/j.atmosres.2018.12.029, 2019.

Mohammed, S.A., Hamouda, M.A., Mahmoud, M.T., Mohamed, M.M., 2020. Interactive comment on "Performance of GPM-IMERG precipitation products under diverse topographical features and multiple-intensity rainfall in an arid region" by Safa A. Mohammed et al. Hydrol. Earth Syst. Sci. Discuss., https://doi.org/10.5194/hess-2019-547-AC3, 2020.
* * *